# The Ovule Number Variation Provides New Insights into Taxa Delimitation in Willows (*Salix* subgen. *Salix*; Salicaceae)

**DOI:** 10.3390/plants12030497

**Published:** 2023-01-21

**Authors:** Alexander M. Marchenko, Yulia A. Kuzovkina

**Affiliations:** 1Russian Park of Water Gardens, Polevaya Str., 12, Klyazma mkr, 141230 Pushkino, Russia; 2Department of Plant Science and Landscape Architecture, University of Connecticut, 1376 Storrs Rd., Storrs, CT 06269, USA

**Keywords:** willow, micromorphology, ovary, capsule, taxonomy, Salicaceae

## Abstract

*Salix babylonica*, *S. alba* and *S. fragilis* are closely related species characterized by the lanceolate, acuminate and serrulate leaves. The boundaries between them are defined by relatively few diagnostic characters, and their identification is not fully solved. Recent studies have demonstrated that the number of ovules present in the ovaries of the willow flower can assist in the identification of the species. The detailed ovule data, characteristic for flowers of each species, *S. babylonica*, *S. alba* and *S. fragilis*, and variation in the number of ovules per ovary were documented using many representatives of these species from various geographic regions. The data included the minimum and maximum number of ovules per valve and per ovary and the percentages of valves with a specific number of ovules in a catkin. Some intermediate genotypes and clusters with similar ovule indexes were observed. The important character for the identification of *S. babylonica* was the presence of valves with 1 or 2 ovules in the ovaries; *S. fragilis* had valves with 3 ovules while *S. alba* had the greater number (4–12).

## 1. Introduction

The genus *Salix*, which comprises approximately 450 species of trees and shrubs, represents a group of considerable taxonomic complexity for several reasons. Considerable individual variability and polymorphism along with phenotypic variability during different developmental stages and habitat conditions limit the diagnostic value of many characters [1,2]. The dioecy excludes the full range of reproductive structures important for identification of an individual plant while the nonconcurrent phenology limits observation of generative and vegetative structures at the same time. In addition, natural tendencies toward hybridization, introgression and allopolyploidy complicate the taxonomy of willows.

Traditionally, willow identification was based on morphological characteristics. Later, cytological, genetic, chemical and ecological distinctions were used to differentiate the species [2]. This was followed by extensive studies in molecular biology, which became helpful in defining species limits [3,4]. Most recent investigations have demonstrated that the number of ovules present in the ovaries of the willow flower can be used to confirm the identification of species and hybrids and, in conjunction with traditional morphological and modern molecular techniques, presents additional evidence to support taxonomic decisions [5].

*Salix babylonica* L. from the section *Subalbae*, *S. alba* L. and *S. fragilis* L. from the section *Salix* according to Skvortsov [1] are closely related species characterized by the lanceolate, acuminate and serrulate leaves. All three species belong to the subgenus *Salix* and are morphologically similar [1]. Skvortsov and Golisheva [6] also described similarities in the leaf anatomy of these three species. According to Fang at al. [7], these three species belong to the section *Salix*.

*Salix babylonica* is a tree 15–18 m in height and up to 80 cm in trunk diameter with a broad spreading crown, slender pendulous or upright branches. The exact limits of the natural range of *S. babylonica* are not known. According to Fang at al. [7], it is widespread throughout China, or, according to Skvortsov [1], it is distributed along river valleys in arid and semiarid regions of central and northern China. *Salix babylonica* is common in cultivation around the world. The northern limit of its successful cultivation in Eurasia is sympatric with the northern margin of commercial peach production, which includes southern England, Belgium, southern Germany, the Czech Republic, Hungary, southern Romania, the Crimea, Caucasus, Uzbekistan, lowlands of Kyrgyzstan, north-east China and the Korean Peninsula [1]. There are many selections and hybrids of *S. babylonica* and many names associated with them. The taxonomy and application of these names have been confusing [8].

*Salix alba* (white willow) and *S. fragilis* (crack willow) are common Eurasian tree species with natural distributions throughout Europe and western and central Asia [1]. Both species belong to the section *Salix* and are morphologically similar, but the delimitation of these species is evident from genetic studies [9,10,11,12,13]. A few studies published the lists of morphological characters useful for distinguishing between *S. alba* and *S. fragilis,* including the most comprehensive treatment by Skvortsov [14]. Still, the boundaries between these species are defined by relatively few diagnostic characters. (The authors continue to use *S. fragilis* as the name for a glabrous crack willow [15] following Skvortsov [1,14]).

Generally, *S. alba* is described as having pubescent stems, leaves and buds while *S. fragilis* is a glabrous willow. However, the degree of the pubescence of *S. alba* varies greatly. “The problem of the identification of pure and hybrid *S. alba* and *S. fragilis* entities is one of the critical matters of systematics of this genus” [16]. Many molecular studies were conducted to clarify definitions of these species [10,11,12,13,16,17,18,19,20,21]. Some of them showed high genetic similarity and suggested a shared ancestry for *S. alba* and *S. fragilis* [12,16,22].

Ovule number constitutes an important taxonomic character for distinguishing between various taxonomic groups. The ovule number is a basic trait that determines the potential fertility of plants with more ovules producing a greater proportion of seeds, thus affecting reproductive success [23,24,25]. The ovule number presents a technological advancement and confers some advantage over the traditional morphological characters. While most of the morphological traits are descriptive with a continuous variation without having a clear diagnostic value, the ovule number has a defined numerical range, is stable and is consistent for the species of *Salix*. Moreover, the ovule count method has a high-resolution power and can indicate the presence of genomes from other species with minimal morphological expression [2,5,15,26,27,28,29,30]. It was recommended that the ovule numbers should be included in the descriptions of new taxa and further taxonomic studies of willows [5].

Previously, Chmelař [26] and Argus [2] listed the ovule index (min–max values) for *S. babylonica* as 4–4 and 2–4, respectively. The index 2–4 implied that there were valves with 1 and 2 ovules per valve and the index 4–4 that there were valves only with 2 ovules per valve (a normally developed non-aberrated willow ovary contains two valves). In addition, some information on the ovule number for *S. alba* and *S. fragilis* was previously reported. Chmelař [26], Valyagina-Malutina [28] and Argus [2] listed the ovule index for *S. alba* as 12–16, 6–10 and 8–9, respectively. For *S. fragilis,* the ovule index was reported as 6–8 by Chmelař [26] and 6–6 by Valyagina-Malutina [28]. Though the ovule ranges were quite different in these reports, especially for *S. alba*, the common trend was that the ovule number was about 6 for *S. fragilis* while *S. alba* had greater values.

No information how these researchers obtained these data was presented. It was not clear how many individuals and from which regions were used to define these ranges. No vouchers were designated in these reports, and thus, corresponding material could not be traced to correlate these findings.

Hence, a study of the ovule number for *S. babylonica, S. fragilis* and *S. alba* was conducted. The objectives were to establish the baselines for the ovule number for these three species and to determine the extent of its variations. Many representatives of *S. babylonica*, *S. fragilis* and *S. alba* from various geographic regions were analyzed. This information is important for defining the species limits.

## 2. Materials and Methods

### 2.1. Plant Materials

For *S. babylonica*, thirty specimens were selected based on their names, which had epithets “babylonica” and “matsudana” (Appendix A). For *S. alba*, *S. fragilis* and their hybrid, seventy-six specimens, collected in various parts of Europe and Asia, were analyzed (Appendix B). The specimens were selected based on their names, which had epithets “alba” and “fragilis”, “alba × fragilis” and “excelsa”. Mostly, specimens identified by prominent salicologists were included into this study. The study also included two other species with lanceolate acuminate and finely serrated leaves—*S. mucronata* Thunb. and *S. tetrasperma* Roxb. *(*one specimen each).

### 2.2. Ovule Count

Willow flowers are arranged into catkins. Each pistillate flower consists of a single pistil with an ovary, floral bract, and nectary (or nectaries). The willow fruit, or a capsule, is formed by the fusion of two carpels. Each capsule contains two valves, in which ovules were counted. Either ripe or unripe capsules were used for ovule count. In ripe capsules, the counts were based on the number of funiculi (from both undeveloped ovules and developed seeds) in the valves. When unripe capsules were used, the counts were made by forcibly opening immature ovaries and counting the number of ovules present in the valves.

For each specimen, the fractions of valves with different numbers of ovules and the number of ovules per ovary (min/often and max/often, “often” means >50%) were documented. The ovule indices were recorded as the minimum–maximum range of ovules per ovary in a catkin (for example, *n* = 10–12) [5,29,30]. These data resulted in the detailed characterization of the ovule distribution and efficient fingerprinting of the genotypes. The ovule number was integrated with some morphological parameters. The presence of trichomes at bract apex, number of nectaries (1 or 2) and pubescence of the ovary were recorded for *S. babylonica*. The presence of trichomes at bract apex and their length were included for *S. alba* and *S. fragilis*, as these characters were previously used to distinguish these species; according to Skvortsov [14], *S. fragilis* has long straight trichomes, which extend by 0.8–2.0 mm beyond the bract margin while trichomes in *S. alba* extend by only 0.2–0.6 mm beyond bract margin.

## 3. Results

The specimens were sorted based on the ovule ranges and arranged as a continuum of genotypes with gradually increasing ovule indexes, starting with the lowest values. For *S. babylonica* (Table 1) the specimens were listed in the descending order of valves with 1 ovule, in parallel to the ascending number of valves with 2 ovules. Starting with Specimen 19, plants had also valves with 3 ovules and were listed with increasing percentages of these valves.

For *S. alba* and *S. fragilis*, the specimens with 2–6 ovules per valve were arranged in Table 2 and the specimens with 6–12 ovules per valve in Table 3. The specimens with similar ovule indexes were separated by bold horizontal borders in Table 1, Table 2 and Table 3.

### 3.1. Salix babylonica

The ovule number in *S. babylonica* represents a continuous variation with following ovule indexes: 2–3, 2–4, 3–4, 4–4, 3–5, 4–5, 4–6, 2–6 and 2–8 (Table 1). A few trends were discerned. Almost all specimens, except Specimen 1, had the majority of valves with 2 ovules—the important character for the identification of *S. babylonica*. Specimen 1 was the only specimen that had most valves with 1 ovule and some valves with 2 ovules. Specimens 2–13 had valves with 1 and 2 ovules. Specimens 14–18 had only valves with 2 ovules. Starting with Specimen 19, there was a progressive increase of valves with 3 ovules/valve, along with the presence of some valves with 1 and 2 ovules. Thus, it was recorded that *S. babylonica* was characterized by the presence of most valves with 2 ovules with a possibility of the occurrence of some valves with 1 and 3 ovules.

Notably, in Specimens 6 and 11 the ovule indexes were different while the percentages of ovules per valve were the same. This is explained by different ovule distribution: Specimen 6 had an arrangement when both valves had 1 ovule, which resulted in the minimum value of 2 ovules pre ovary in the index 2–4. This arrangement did not occur in Specimen 11, which had no ovaries where both valves had 1 ovule, resulting in the minimum value of 3 (ovule index 3–4).

Starting with Specimen 19, the occurrence of valves with 3 ovules/valve resulted in the increase of the maximum value in the ovule indexes (3–5, 4–5, etc.). A larger fraction of valves with 3 ovules in Specimen 30 and the presence of valves with 4 ovules expanded the ovule index to a broader range of 2–8.

Some clusters with similar ovule indexes were observed.

#### 3.1.1. *S. babylonica* var. *sacramenta* Marchenko et Kuzovkina, ined.

Specimen 1 is listed as *S. babylonica* var. *sacramenta*, represented by a cultivated plant that differed from other specimens by having most valves (61%) with 1 ovule and the ovule index 2–3. The name *S. babylonica* var. *sacramenta* was not validly published. The origin of this taxon was not known as there were confusing reports about its introduction. The plant under this name was introduced to Argentina in 1928, either from Russia through USA [31] or from Switzerland [8]. Ragonese and Alberti [32] noted that this clone was introduced to Argentina as *S. babylonica* var. *sacramenta* Hortus from Botanical Garden of the University of Copenhagen, Denmark, where it was in cultivation since 1893 from cuttings received from Moscow. They noted that this plant was apparently never described, and there was no possibility to find bibliographic information about it. Krussmann [33] described *S. babylonica* ‘Sacramento’ as a relatively little-known clone with a less pronounced weeping habit, open crown and large leaves.

We have studied *Salix babylonica* L. var. *sacramenta* cultivated in Argentina. It is a tree with slightly pendulous branchlets (Figure 1). The green-gray bark of branches and branchlets retains its color on the side exposed to the sun. Branchlets consist of a few orders of shoots, which develop during a growing season. The leaves have sharp serration and much broader blades are distinguishable from those in other varieties of *S. babylonica* (Figure 2). Other important characteristics include small stipules, in the shape of small glands, almost invisible on pubescent stems. The buds are ovoid and flattened, partially pubescent on the upper half. The outermost cataphylls inside the buds are mostly glabrous, with hair only above the midrib and along the margins. Catkins are short and densely flowered. Flower bracts are wide rounded, often with uneven edges. Pistillate flowers are borne on a short stipe; pistils have a short style, often with 1 nectary. The epithet *sacramenta* derives from the name under which this plant was introduced to Argentina.

There were some similarities between Specimen 1 and Specimens 2–6, which had a greater number of valves with 1 ovule compared to other specimens. While the origin of Specimen 2, representing a cultivated plant, was unknown, Specimens 3–5 were collected by Skvortsov in Yunnan province in southern China*,* suggesting that it was possible that *S. babylonica* var. *sacramenta* also originated from the south of China. Notably, Skvortsov placed Specimens 3–5 into a folder called *S. babylonica,* though the individual specimens were identified only as “*Salix*”. These specimens had the unusually thin and short annual branchlets, and it was possible that Skvortsov was uncertain about their identity. A similar branching pattern was observed in *S. babylonica* var. *sacramenta*, which resulted from the growth of a few cycles of branchlets during a growing season. At the end of each cycle, the tip of the branchlet ceased growth and died; this event was followed by the development of the nearest to the tip axillary bud, which grew into a new shoot. The new shoot pushed the dead tip aside and continued its growth along the previous axis of the branchlet. This process continued throughout the season resulting in a few orders of branchlets forming long and thin stems.

Specimens 2–6, characterized by the ovule index 2–4, were likely the intermediate genotypes between *S. babylonica* var. *sacramenta* and *S. babylonica* ‘Babylon’. Notably, Skvortsov annotated Specimen 6, stating that it was very similar to his samples from China collected in 1998 (Specimens 3–5), and the data on the ovule number confirmed this.

#### 3.1.2. *S. babylonica* ‘Babylon’

Specimens 7–10 had the ovule index 3–4 and similar morphological characters. Specimen 10 of *S. babylonica* ‘Babylon’ was studied by Santamour and McArdle [8] and was confirmed to be the pure species based on the previously conducted chemical analysis.

*Salix babylonica* ‘Babylon’ is a graceful tree with long pendulous slender branches and brown, often purple on the upper side of branchlets (Figure 3). The outermost cataphylls inside the buds are mostly glabrous with hair only above the midrib and along the margins, which is different from *S*. *babylonica* var. *matsudana* H.Ohashi & Yonek (Figure 4). The cultivar name ‘Babylon’ was proposed by Santamour McArdle [8] for the female clone, which possibly provided the basis for the original species description by Linnaeus. They described it as a highly atypical selection from the species that had been introduced along the ancient trade routes through southwest Asia, to the Near East, and to Europe around 1730.

#### 3.1.3. *S. babylonica* var. *matsudana*

This group includes Specimens 11–24, most of which were previously identified as *S. matsudana.* Historically, the two binominals—*S. babylonica* vs. *S. matsudana*—were used interchangeably by various researchers. The ovary pubescence and the number of nectaries were usually used to distinguish these two species. Skvortsov [1] and Argus [2] pointed to the inconsistency of the ovary pubescence. A similar trend was also documented here; while most of the specimens had glabrous ovaries, Specimen 18 had pubescent ovaries, and Specimen 19 had both pubescent and glabrous ovaries. Further, Skvortsov [1] and Argus [2] noted that *S. babylonica* had one nectary while *S. matsudana* had two. However, all specimens identified as *S. matsudana* in Table 1 had one nectary except for *S. matsudana* ‘Umbraculifera’ (Specimen 23), in which only 30% of the flowers had two nectaries. Moreover, in some specimens, flowers had 2 nectaries at the base of the catkins, and the rest of the flowers had only one nectary (Specimens 19 and 23). The number of nectaries varied not only among different specimens of the same species, but also in different branches or catkins of the same specimens.

However, a consistent character was observed, which can be used to distinguish between these two taxa. In all specimens of *S. matsudana*, including ’Annularis’ (*S. babylonica* ’Crispa’)*,* ‘Tortuosa’, and ‘Umbraculifera’*,* the outermost cataphylls inside the buds had dense and long trichomes; while in the specimens of *S. babylonica*, the outermost cataphylls were mostly glabrous, with hair only above the midrib and along the margins. The taxonomic importance of this character for species differentiation should be further assessed. Noteworthy, Skvortsov [1] used a similar character—pubescence along the edges of the outermost cataphylls—to differentiate two related species *S. pentandra* L. and *S. pseudopentandra* Flod.

Some differences between *S. babylonica* and *S. matsudana* were observed in cultivation as well. Generally*,* cultivars of *S. matsudana* are drought tolerant and tough plants [34]. *Salix babylonica* and its cultivars were relatively sensitive to winter cold, had limited frost hardiness, and almost never survived north of the USDA Zone 7 without significant injuries. *Salix matsudana* and its cultivars were hardier in northern climates and could be grown to at least USDA Zone 4a with winter temperatures at −34 °C to −32 °C [35].

The ovule index of specimens previously identified as *S. matsudana* was 3–5 and included the ranges of 3–4, 4–4, 3–5, 4–5 and 4–6. Most of the specimens had almost all valves with 2 ovules, a few specimens (19–22) had very few, almost “accidental”, valves with 3 ovules while specimens 23 and 24 had even greater number of valves with 3 ovules.

The recent discussion on the taxonomic position of *S. matsudana* by Ohashi and Yonekura [36] recognized a new variety *S. babylonica* var. *matsudana* (Koidz.) H.Ohashi & Yonek. The following specifications, including the pubescent outermost cataphylls and 3–6 ovules per ovary, as described above, should be added to accurately define *S. babylonica* var. *matsudana*.

The three cultivars—‘Annularis’ with leaves curved into rings (Specimen 11), *‘*Tortuosa’ with irregularly contorted branches and leaves (Specimen 12–15, 21, 22), and *‘*Umbraculifera’ with a dense subglobose crown (Specimen 23 and 24)—appeared to belong to var. *matsudana* based on the presence of the pubescent cataphylls. Cultivars *‘*Annularis’ and ’Tortuosa’ represent the products of ancient Chinese selection according to Skvortsov [1]).

‘Annularis’ was reported to be more cold hardy than *S. babylonica*, which is in accord with genotypes of *S. babylonica* var. *matsudana.*

The specimens of ’Tortuosa’ had the intermittent distribution in Table 1; there were specimens with 1 and 2 ovules per valve (Specimens 12 and 13); with only 2 (Specimens 14 and 15); with 1, 2, and 3 (Specimen 21); and with 2 and 3 (Specimen 22) ovules per valve, though the greater majority of valves with 2 ovules and 4–4 index were observed. There were noticeably fewer valves with 1 ovule compared with *S. babylonica*. ‘Tortuosa’ is characterized by the ovule index 3–5; most ovaries were with 4 ovules.

Specimens 23 and 24 identified as *S. matsudana ‘*Umbraculifera’ had no valves with 1 ovule, and a much larger (19–24%) proportion of valves with 3 ovules. While the pure *S. babylonica* is defined by the prevalence of the valves with 2 ovules, the occurrence of a larger number of valves with 3 ovules indicates its relation to *S. fragilis* (the pure *S. fragilis* has all valves with 3 ovules). Noteworthy, the shape of the crown of *S. matsudana* ‘Umbraculifera’, which resemble *S. fragilis* ‘Bullata’, enabled confidently distinguish this plant (Figure 5). Blackening buds in *S. matsudana* ‘Umbraculifera’, was another distinguishing character indicating its close relationship with *S. fragilis*.

The color of thin branches and branchlets of ‘Umbraculifera’ ranges from yellow-greenish to purple-brownish (the upper side of the branches exposed to the sun gets tanned, gaining purple color). Mature branches and leaves are glabrous. Buds blunt at the apex (*S. fragilis* ‘Bullata’ has acuminate buds). Outermost cataphylls inside the buds densely pubescent as in *S. babylonica* var. *matsudana* (glabrous *S. fragilis* ‘Bullata’) (Figure 5). Young leaves are reddish; unfolding leaves are densely pubescent on the upper side, rarely on the lower side and mainly along the central vein. Stipules are large, with rolled up edges and sparse hairs. Stipule attachment point, as in *S. fragilis* ‘Bullata’, is on the side of the lower third part of the stipule (in *S. babylonica* var. *babylon*—at the base) (Figure 6). Bracts glabrous is acuminate. Ovaries sessile with a very short stipe. Nectary occurred as one or rarely two.

Based on the morphological characters and ovule number, this taxon could be a hybrid of *S. babylonica* var. *matsudana* (3–5 ovules/ovary) and *S. fragilis* ‘Bullata’ (6–6 ovules/ovary). Such a hybrid would have the predicted ovule index of 4–6. However, it is more sensible to rather consider this plant as the intermediate genotype between *S. babylonica* var. *matsudana* and *S. fragilis.*

As evident from the labels, Specimen 24 collected in 1984 at Arnold Arboretum was procured from Morton Arboretum in 1960, and thus it likely represented the same clone as Specimen 23, the catkins from a live specimen from Morton Arboretum, which were procured in 2017. Thus, Specimens 23 and 24 collected 33 years apart had very similar ovule counts: 85% and 78% of valves with 2 ovules, 15% and 22% of valves with 3 ovules, correspondingly, and ovary index 4–6. The small differences in the percentages of valves between two specimens, representing the same clone but grown under different conditions and collected at a large time interval, indicated the stability of this trait.

Specimen 25 collected and identified as *S. babylonica* by the Swedish researcher S.J. Enander in 1913 in Transbaikalia, north of China, had a large proportion (20%) of valves with 3 ovules. Its identification needs clarification as its leaves were similar to *S. fragilis* but the crown shape was not evident from the herbarium specimens. This specimen was notable due to its northern occurrence, which is unusual for *S. babylonica*.

Specimens 27 and 28 from the north of China were less weeping as was evident from the herbarium specimens. Specimens 27 was cited by Skvortsov [1]; p.121 as a remarkable specimen representing a tree up to 20 m tall and more than 1.75 m in diameter found in the oases of A-la Shan (Ho-lan Shan) in 1909 by S. Chetyrkin. These specimens had no valves with 1 ovule and a few valves with 3 ovules.

#### 3.1.4. *S. babylonica-fragilis*—Group (The Groups Were Originally Designated as Varieties by A.M.; such Designations Were not in Accord with the ICBN and Were Replaced with the Non-Nomenclatural Designations Such as Groups

Specimen 29, previously identified as *S. fragilis* by Skvortsov, had a few valves (10%) with 1 ovule, most (61%) with 2 ovules, and some (29%) with 3 ovules. While the presence of valves with 1 and 2 ovules is typical for *S. babylonica*, the greater presence of valves with 3 ovules, typical for *S. fragilis*, indicated greater involvement of that species. The *S. babylonica-fragilis*—group is characterized by the ovule index 2–6. It has valves with 1, 2, and 3 ovules with most valves with 2 ovules.

#### 3.1.5. *S. babylonica-alba*—Group

Specimen 30, which had very few valves (3%) with 1 ovule, some (24%) with 2 ovules, most (56%) with 3 ovules, and some (17%) with 4 ovules was placed at the end of Table 1. While the presence of valves with 1 and 2 ovules connects this specimen to *S. babylonica*, the presence of valves with 3 ovules indicates the involvement of *S. fragilis* while the presence of valves with 4 ovules indicates the involvement of *S. alba*. Thus, this specimen is considered to be the intermediate genotype (Table 2). The *S. babylonica-alba*—group has ovule index 2–8 and valves with 1, 2, 3, and 4 ovules.

#### 3.1.6. *Salix babylonica dolorosa*

Another obscure taxon “*S. babylonica dolorosa*” was analyzed to clarify its affiliation. Santamour and McArdle [8] mentioned that the name “dolorosa” was used variously as *S*. *dolorosa* Hort., or in the synonymy with *S. babylonica*, or as a variety *S. babylonica dolorosa* Hort. *Salix babylonica* var*. dolorosa* Rowen ex Rowlee was suggested to be a variety of the species by Bailey [37]. It was listed as a plant with leaves glaucous underneath, with the common name Wisconsin weeping willow, with a note that this plant is hardy farther north. Bailey [38] referred to it as a synonym of *S. blanda*, a suggested hybrid of *S. babylonica* and *S. fragilis*, and this placement was followed by Rehder [39].

The analysis of the ovule number of the herbarium specimen of *S. babylonica* var*. dolorosa* (W.L.G. Edson; 0-711; Highland Park, Rochester, NY, US, collected 8 June 1918, A) indicated the following: 1% of valves with 2 ovules, 88% of valves with 3 ovules, 10% of valves with 4 ovules, and 1% of valves with 5 ovules based on the analysis of 64 ovaries. Its ovule index was 5–8, and a very similar distribution of valves with various ovule number corresponded to *S. ×pendulina* Wender.

The ovule number in *S. alba*, *S. fragilis*, or *S. alba* × *S. fragilis* is presented in Table 2 and Table 3. All specimens had various proportions of valves with a different number of ovules. At the beginning of Table 2 the specimens were listed in descending order of valves with 2 ovules, in parallel to the ascending number of valves with 3 ovules. Some of these specimens also had small fractions of valves with 4 or 5 ovules. A gradual transition toward the specimens with greater than 3 ovule per valve was observed toward the end of Table 2 and throughout Table 3. In parallel to the increasing number of ovules per valve, the ovule indexes were also gradually increasing from 4–6 to 20–24. The two groups were distinguishable.

### 3.2. S. fragilis

Specimens 1–32 had most valves with 3 ovules (Table 2). The ovule indices varied from 4–6 to 6–9 and included the intermediate ranges of 4–7, 4–9, 5–6, 6–6, 5–8, 6–7, and 6–8. The distribution of valves with various ovule number varied among specimens.

#### 3.2.1. *S. fragilis-babylonica*—Group

Specimens 1–5 had most valves with 3 ovules, typical for *S. fragilis,* and few valves with 2 ovules, typical for *S. babylonica*. This subset with the characteristic ovule index of 4–6 was recognized as the intermediate between *S. fragilis* and *S. babylonica* and designated as the *S. fragilis-babylonica*—group, characterized by the ovule index 4–6. It has valves with 2 and 3 ovules with most valves with 3 ovules.

The analysis of the ovule number in *S. babylonica* described above also identified some specimens as the intermediate variants from *S. babylonica* to *S. fragilis.* Those specimens designated as belonging to the *S. babylonica-fragilis*—group had greater involvement of *S. babylonica,* as was evidenced by most valves with 1 or 2 ovules, along with some valves with 3 ovules, and the characteristic ovule index 2–6.

#### 3.2.2. The Pure *S. fragilis*

Specimen 6 had ovule index 6–6, which means that all valves in the ovaries of catkins had 3 ovules. The previously documented ovule index for the pure *S. fragilis*, which was obtained through the study of the specimens collected in natural areas [15,28] and predicted through the analysis of its hybrids [29], was 6–6. Notably, Specimen 6 was the only specimen that matched the previously reported ovule index for the pure *S. fragilis*.

#### 3.2.3. *S. fragilis-alba*—Group

Specimens 7–12 had a small proportion of valves with 2 ovules, most valves with 3 ovules along with some valves with 4 and 5 valves. Specimens 13–22 had most valves with 3 ovules and some valves with 4 ovules, while Specimens 23–32 had also valves with 5 ovules, which are associated with *S. alba*. This group having most valves with 3 ovules, typical for *S. fragilis*, was recognized as the intermediate between *S. fragilis* and *S. alba* and designated as *S. fragilis-alba*—group with the ovule index 6–9. It has valves with 3, 4 and 5 ovules with most valves with 3 ovules.

### 3.3. S. alba

#### 3.3.1. *S. alba-fragilis*—Group

Starting with Specimen 33 and continuing through Specimen 50, there was a noticeable transition to the genotypes with less than 50% of valves with 3 ovules, in parallel with the greater number of valves with 4 and 5 ovules, typical for *S. alba*. This group was designated as *S. alba-fragilis* with the ovule index 6–12 (Table 2). It has valves with 3, 4, 5, and 6 ovules with less than 50% of valves with 3 ovules.

The most common ovule index was 6–10 (a few specimens had the ovule index 7–9, 7–10, 8–10, and 8–11, which were within 6–12 range). The lower value of 6 in this index meant that in these specimens there were some ovaries were both valves had 3 ovules, as in the pure *S. fragilis*; while the greater upper values in the ovule indexes meant that there were also ovaries with 4, 5, 6 ovules per valve. Specimen 49, previously identified as *S. euxina* I.V.Belyaeva [40], belongs to this group [15].

#### 3.3.2. The Pure *S. alba*

Specimens 51–62 had no valves with 3 ovules (associated with *S. fragilis*), a noticeably higher proportion of valves with 4, 5, and 6 ovules. Specimen 56, first identified by various botanists as pure *S. fragilis*, was proposed as a new type for that species [41], and later, annotated as the epitype of *S. euxina* in 2019, belongs to this group [15].

Staring with Specimen 63 there were no specimens that had valves with 4 ovules. Starting with Specimen 67 (Table 3), there were no specimens with 5 ovules/valve. The specimens with greater proportion of valves with even larger number of ovules were recorded throughout Table 3.

#### 3.3.3. *S. alba-excelsa*—Group

In Specimens 70–74 the ovule indexes reached the ranges 15–17, 15–18, 16–21, 18–20, and 18–21. This group with high ovule indexes was designated as *S. alba-excelsa*—group with the ovule index of 15–21 and valves with 7, 8, 9, 10, 11, and 12 ovules (Table 3). This group included specimens previously identified by Skvortsov as *S. excelsa* J.F.Gmel.*,* a species occurring in Iran, Afghanistan, and Middle Asia, which he considered to be closely-related to *S. alba* [1]

### 3.4. Specimen Identifications

When previous identifications of the specimens of *S. alba*, *S. fragilis* and their hybrid were checked against the ovule-based data, the variegation in column 1, related to color-coding of the species’ names, was apparent throughout Table 2. Instead of the expected clustering of the specimens belonging to the same species, their scattered distributions indicated the discrepancies between the placement of the specimens in the tables based on the ovule data, and the previous identifications based on the traditional morphologically characters.

Specimens 1–50, which included specimens with 3 ovules per valves appeared to be the most difficult group for identification. Contrary, in almost all specimens without 3 ovules per valve (starting from Specimen 51 in Table 2 and all specimens in Table 3), the ovule number corroborated the specimen identifications as *S. alba* by different authors. This fact meant that in these specimens there was a strong expression of morphological traits of *S. alba*, which unambiguously suggested its recognition.

At the other end, all specimens previously identified as *S. fragilis* (except Specimen 75) had valves with 3 ovules. It is likely that even a small fraction of the valves with 3 ovules, resulted in the expression of the morphological traits typical for *S. fragilis*. However, in some specimens in the group where most valves had 3 ovules, the presence of valves with a different number of ovules apparently blurred their recognition as *S. fragilis*.

Notably, in some cases the specimens with similar ovule number were identified interchangeably as different species by the same authors. For example, Specimens 13, 14, 16, 18 had similar proportions of valves with 3 and 4 ovules but were identified either as *S. fragilis* (Specimens 14 and 18) or as *S. alba* (Specimens 13 and 16) by Skvortsov, who was often considered to be one of the most experienced researchers of *Salix*. Based on the ovule number, which included most valves with 3 ovules, all specimens should belong to *S. fragilis* (*S. fragilis-alba*—group).

As to the analysis of other morphological characters, many specimens had long bract hair in the *S. fragilis-*group, fewer in the *S. alba-*group in the middle of Table 2, and no specimens with long bract hair in the *S. alba-*group at the end of Table 2 and throughout Table 3. All specimens identified as *S. fragilis* by Skvortsov had bracts with long trichomes at the apexes. Apparently, Skvortsov considered this character as diagnostic for *S. fragilis*, though some of these specimens with the long bract hair were not confirmed to be *S. fragilis* by the ovule number. For example, Specimen 33 and 41, identified by Skvortsov as *S. fragilis*, had long trichomes at the bract apex, but, based on the presence of the most valves with 4 ovules and ovule indexes 6–10, should rather belong to *S. alba.*

### 3.5. Salix tetrasperma and S. mucronata

The ovule number for two other species with lanceolate acuminate and serrulate leaves similar to *S. babylonica, S. alba, *and *S. fragilis*—*S. tetrasperma* (Indian willow) from southern and southeastern Asia and *S. mucronata* (Cape willow, or safsaf willow) from South Africa were analyzed using the limited number of specimens. *Salix tetrasperma* had 100% of the valves with 2 ovules (ovule index 4–4, based on the analysis of 52 ovaries from Specimen NA0556465) while *S. mucronata* had 100% of valves with 6 ovules (ovule index 12–12, based on the analysis of 23 ovaries from Specimen NA0556301). *Salix tetrasperma* grows in the climatic conditions similar to the conditions of *S. babylonica* and has the identical ovule index. *Salix mucronata,* which naturally occurs in harsher climatic conditions and is adapted to colder and drier, had a greater ovule index.

Another species, *S. humboldtiana* Willd. From the section *Humboldtianae*, is often cultivated in South America (Argentina and Chile) and Mediterranean region of Europe. *S. humboldtiana* is also superficially similar to *S. babylonica*, with which it is occasionally con-fused. However, it is easily distinguishable from *S. babylonica* by much greater ovule number—20–24 [5].

## 4. Discussion

The studied specimens of *S. babylonica, S. fragilis*, or *S. alba* were from various parts of Europe and Asia, providing a broad geographic context for the ovule variations in these species. While they are closely related species, the boundaries between them are defined by relatively few diagnostic characters, and their identification is still not fully solved as was evidenced by this study. The ovule analyses demonstrated that these species can be distinguished by the ovule number. It was concluded that *S. babylonica* had most valves with 1 or 2 ovules, *S. fragilis* with 3 ovules while *S. alba* had the most valves with the higher number (4, 5, and up to 12) of ovules. In addition, various combinations of the smaller fractions of valves with 3 ovules (typical for *S. fragilis*) and 4 ovules (typical for *S. alba*) were recorded in the ovaries of many genotypes of *S. babylonica*. Some specimens had higher ovule indexes than previously reported (2–4 or 4–4) for *S. babylonica*—as 3–5, 4–5, 4–6, 2–6, and 2–8.

The occurrence of valves with a larger number of ovules than typical for *S. babylonica* 1 and 2 ovule/valve indicated the recombination of genomes either through introgression or hybridization. The presence of valves with 3 and 4 ovules indicated the rearrangement of genetic material with some progression to *S. fragilis* and *S. alba*. It was possible that the evolution progressed from *S. babylonica* to a morphologically similar *S. alba* through the intermediate *S. fragilis*, the two species which belong to the European–West Asian section *Salix.*

The studied specimens of *S. fragilis* had lower levels of variation of the ovule number, ranging from 2 to 5, compared to S. alba, the specimens of which contained a broader range of valves with various ovule number, ranging from 3 to 12.

The specimens which belong to *S. fragilis* had the majority valves with 3 ovules, and the specimens “*S. alba*—group” had the majority valves with more than 3 ovules.

When many genotypes of these species were placed in a sequence according to the ovule number, they represented a large continuum of individuals with no clear borders. A gradual transition from *S. babylonica* to *S. fragilis* and then to *S. alba* among studied genotypes was observed as was evidenced by the appearance of valves with larger ovule numbers. The presence of valves with three ovules in certain genotypes of *S. babylonica* indicated their closeness with *S. fragilis* while the presence of valves with four or five ovules indicated their closeness with *S. alba*.

There were many intermediate specimens with various assortments of valves associated with either *S. babylonica, S. fragilis* or *S. alba*. The intermediate genotypes of *S. babylonica* had most valves with 1 or 2 ovules per valve, along with some valves with 3 and 4 ovules. The intermediate genotypes of *S. fragilis* had most valves with 3 ovules per valve, along with some valves with 2, 4, or 5 ovules. The intermediate genotypes of *S. alba* had most valves with 4 or more ovules along with some valves with 3 ovules. The presence of the intermediate forms demonstrated that the boundaries between species were not strictly defined, as would not be expected from the nature. It is better to differentiate them as the intermediate taxa rather than hybrids.

Thus, the rearrangement or recombination of valves with various ovule number was recorded, which probably took place during the evolutionary processes in various climatic conditions. A gradual from 1, 2 to 3, and then to 4, 5 and greater number of ovules/valve between closely related species was observed. This change can be interpreted as the transition from *S. babylonica* to *S. alba* through *S. fragilis* (Figure 7), which should be further confirmed by the phylogenetic data.

A notable Specimen 5 was included into Table 1. This specimen from Iran was at first named as *S. aegyptiaca* by Saberti and later identified as *S. babylonica* by Skvortsov. It had almost all valves with 3 ovules, a very low number of valves (1%) with 2 ovules, and the ovule index 5–6. The prevalence of valves with 3 ovules indicated that this specimen belongs to *S. fragilis* but was misidentified by Skvortsov as *S. babylonica.* This fact points out the morphological similarity of *S. babylonica* and *S. fragilis* corroborating a close relationship of these species. Interestingly, *S. matsudana* Koidz. was distinguished from *S. fragilis* by Koidzumi [36], which also suggests a close connection between these taxa.

Throughout evolutionary history, plants have developed the reliable reproductive mechanisms insuring their sustained survival across generations. Seed production is a primary means of plant propagation, and the number of ovules correlates to the number of future seeds, which directly affect the survival. Evolutionarily, *Salix* probably arose in the warm temperate or subtropical regions of eastern Asia [1]. It is possible that the evolution of the number of ovules per ovary has progressed toward its increasing. *Salix babylonica* var. *sacramenta* and other specimens from Southern China were probably the most ancient representatives in this group. In the more favorable conditions in the south having one ovule/valve or one seed/valve was sufficient for species survival. In the temperate continental climate, willows also produced a few ovules. The gradual increase of the ovule number could have been observed as species of *Salix* extended to the north and south from their optimal environments. For example, the pure *S. fragilis* from Bulgaria has 3 ovules/valve while a larger number of ovules/valve in *S. alba* had evolutionary advantages in colder climates or in hot and dry environments where the greater number of ovules corresponds to the greater number of potential seeds, which can survive in harsh conditions.

## 5. Conclusions

The number of ovules present in the ovary of the normally developed willow flower is a stable trait that can be used to confirm species identification when used in combination with morphological and molecular data. The genetic structure of *Salix babylonica*, *S. alba*, and *S. fragilis* appeared to be complex as can be seen from the existence of many individuals with various proportion of valves with a different number of ovules. As more representatives of these species are analyzed in the future, the more adequately their genetic diversity is captured, resulting in more comprehensive understanding of the species delimitations.

## Figures and Tables

**Figure 1 plants-12-00497-f001:**
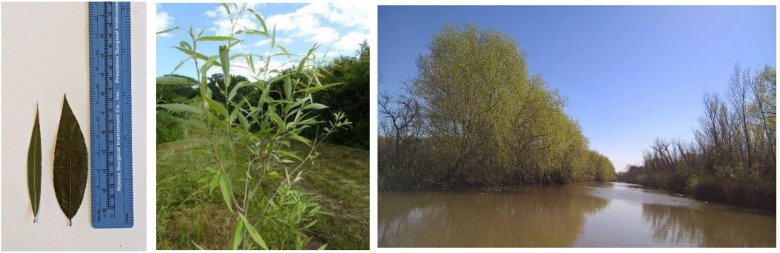
(**Left**) Narrow-lanceolate leaves of *S. babylonica* ’Babylon’ (**left**) and lanceolate leaves of *S. babylonica* var. *sacramenta* (**right**). (**Center**) The green-gray bark of *S. babylonica* var. *sacramenta* retains its color even on the side exposed to the sun. Right: *S. babylonica* var. *sacramenta* in the Paraná Delta region, Argentina (photo by T. Cerrillo).

**Figure 2 plants-12-00497-f002:**
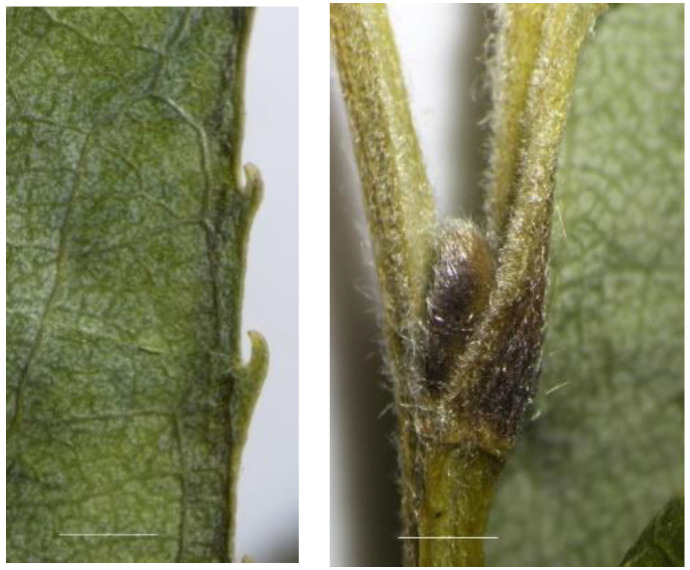
*Salix babylonica* var. *sacramenta*. Leaf margin (**left**). Branchlet with the bud (**right**).

**Figure 3 plants-12-00497-f003:**
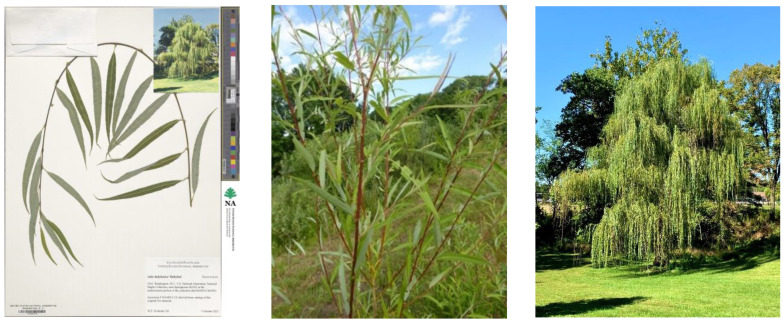
Herbarium specimen of *S. babylonica* ‘Babylon’ (NA0263746) of the live specimen NA44011 (**left**). The green-brown bark of branches of *S. babylonica* ‘Babylon’ NA44011 growing at Storrs, Connecticut, US, gets tanned, gaining purple color on the upper side exposed to the sun (**center**). *S. babylonica* ‘Babylon’ NA44011 growing at the National Arboretum, Washington, US, displays its strongly pendulous habit (**right**) (photo by H. Svoboda).

**Figure 4 plants-12-00497-f004:**
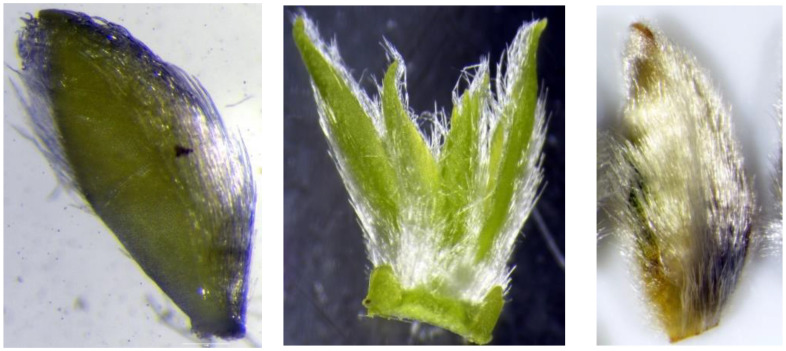
Outermost cataphylls of *S. babylonica* var. *sacramenta* (**left**), *S. babylonica* ’Babylon’ (**center**) and *S. babylonica* L. var. *matsudana* ‘Umbraculifera’ (**right**) after the bud scales were removed. In both taxa of *S. babylonica*—var. *sacramenta* and ’Babylon’—hairs were located only above the central vein and along the edge so that the part of the cataphyll blade lying between the central vein and the edge, remained glabrous, also there were no hairs on its top. In the cataphylls of *S. matsudana* ‘Umbraculifera’, the abaxial side is densely and evenly pubescent with long hairs.

**Figure 5 plants-12-00497-f005:**
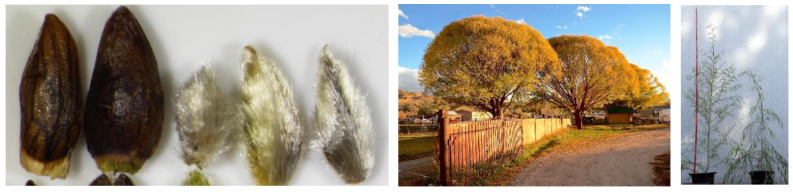
Bud scales and pubescent outermost cataphylls of *S. babylonica* var. *matsudana* ‘Umbraculifera’ (**left**). A unique dense subglobose crown of *S. babylonica* var. *matsudana* ‘Umbraculifera’ (**center**) (photos by M. Dodge). Upright habit of *S. babylonica* var. *matsudana* ‘Umbraculifera’ (**left**) and pendulous habit of *S. babylonica* ’Babylon’ (**right**) visible during their first year of growth (**right**).

**Figure 6 plants-12-00497-f006:**
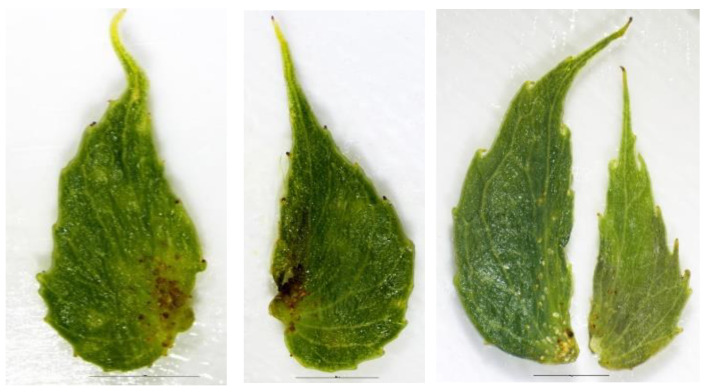
Stipules of *S. babylonica* var. *matsudana* ‘Umbraculifera’ (**left**, **center**) and *S. babylonica* ’Babylon’ (**right**).

**Figure 7 plants-12-00497-f007:**
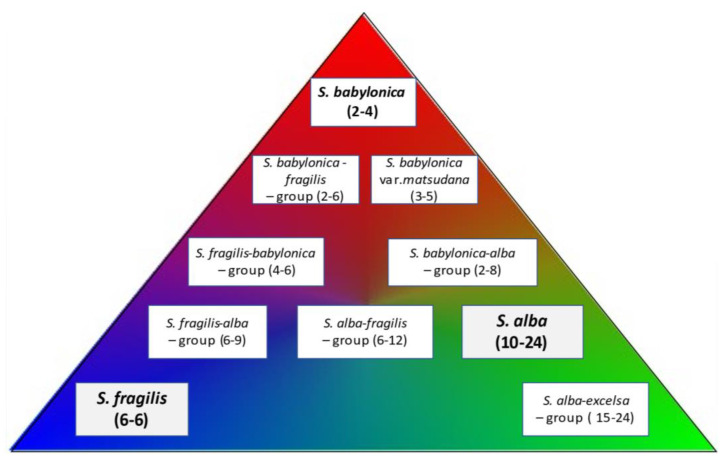
Diagrammatic representation of the variation of the ovule number in *S. babylonica—S. fragilis—S. alba* complex: the pure species and intermediate variants.

**Table 1 plants-12-00497-t001:** The ovule data for *S. babylonica*. Each capsule in *Salix* has 2 valves and the presented data include the percentage of valves in a catkin with different numbers of ovules. Additional characteristics are included: the number of ovules per ovary (min/often and max/often), the ovule indices (the minimum and maximum number of ovules per ovary), presence of trichomes at bract apex (A—bracts abscised; N—none; L—long; S—short), number of nectaries, ovary pubescence (G—glabrous; P—pubescent) and number of studied ovaries.

No	Specimen	Percentage of Valves with Each Ovule Number				No. of Ovules per Ovary		Ovule Index	Trichomes	Nectaries	Ovary	No of Ovaries
		1	2	3	4	Min/ Often	Max/ Often					
1	*S. babylonica* var. *sacramenta* ‘Soveny Americano’	61	39			2/2	3/3	2–3	A	1	G	50
2	*S. babylonica*	28	72			2/3	4/4	2–4	A	1	G	68
3	*Salix*—specimen No. 1 in the folder “*S. babylonica*”	17	83			2/3	4/4	2–4	A	1	G	58
4	*Salix—*specimen No. 2 in the folder “*S. babylonica*.”	16	84			2/3	4/4	2–4	A	2	G	50
5	*Salix*—specimen No. 3 in the folder “*S. babylonica*”	9	91			2/3	4/4	2–4	A	2	G	92
6	*S. babylonica*	8	92			2/3	4/4	2–4	A	1	G	56
7	*S. babylonica*	5	95			3/4	4/4	3–4	A	2	G, P	130
8	*S. babylonica*	4	96			3/4	4/4	3–4	A	1	G	60
9	*S. babylonica*	1	99			3/4	4/4	3–4	A	1	P	58
10	*S. babylonica* ‘Babylon’	1	99			3/4	4/4	3–4	A	1	G	119
11	*S. babylonica* var*. annularis*	8	92			3/3	4/4	3–4	A	1	P at base	109
12	*S. matsudana* f. *tortuosa*	4	96			3/3	4/4	3–4	A	1	G	56
13	*S. babylonica* ‘Tortuosa’	1	99			3/4	4/4	3–4	A	1	G	46
14	*S. matsudana* ‘Tortuosa’	0	100			4/4	4/4	4–4	A	1	G	52
15	*S. matsudana* var. *tortuosa*	0	100			4/4	4/4	4–4	A	1	G	55
16	*S. matsudana*	0	100			4/4	4/4	4–4	A	1	G	32
17	*S. matsudana*	0	100			4/4	4/4	4–4	A	1	G	65
18	*S. matsudana*f. *pendula*	0	100			4/4	4/4	4–4	A	1	P	56
19	*S. babylonica*	3	96	1		3/4	5/4	3–5	A	1, 2	G, P	91
20	*S. matsudana*	1	98	1		3/4	5/4	3–5	A	1	G	105
21	*S. matsudana* var. *tortuosa*	1	93	6		3/4	5/5	3–5	A	1	G	97
22	*S. babylonica* ‘Tortuosa	0	99	1		4/4	5/4	4–5	A	1	G	56
23	*S. matsudana ‘*Umbraculifera’	0	85	15		4/4	6/5	4–6	A	1, 2	G	35
24	*S. matsudana* ‘Umbraculifera’	0	78	22		4/4	6/5	4–6	A	1	G	45
25	*S. babylonica*	0	80	20		4/4	6/5	4–6	A	1	G	20
26	*S. babylonica*	0	99	1		4/4	5/4	4–5	A	1, 2	G, P	132
27	*S. babylonica*	0	98	2		4/4	5/4	4–5	A	1	G	191
28	*S. babylonica*	0	97	3		4/4	5/4	4–5	A	1	G	57
29	*S. fragilis*	10	61	29		2/4	6/5	2–6	L	1	G	38
30	*S. fragilis*	3	24	56	17	2/4	8/6	2–8	L	1	G	38

**Table 2 plants-12-00497-t002:** The ovule data for *Salix alba*, *S. fragilis* and *S. alba* × *S. fragilis* with 1 to 6 ovule/valve. Each capsule in *Salix* has 2 valves and the presented data include the percentage of valves in a catkin with different numbers of ovules. The additional characteristics are included: the number of ovules per ovary (min/often and max/often), the ovule indices (the minimum and maximum number of ovules per ovary), presence of trichomes at bract apex (A—bracts abscised; L—long; N—none; S—short), and number of studied ovaries. The entries with the specimens’ numbers (column 1) were colored according to their names based on their original identification—blue for *S. alba*, yellow for *S. fragilis*, red for *S. alba* × *S. fragilis*—for visual presentation of the species’ distribution throughout the tables.

No	Specimen	Percentage of Valves with Each Ovule Number					No. of Ovules per Ovary		Ovule Index	Trichomes	No of Ovaries
		2	3	4	5	6	Min/ Often	Max/ Often			
1	*S. fragilis* f. *latifolia*	33	67	0	0	0	4/4	6/6	4–6	L	70
2	*S. alba*	27	73	0	0	0	4/5	6/6	4–6	L	44
3	*S. alba × S. fragilis*	12	88	0	0	0	5/5	6/6	5–6	L	82
4	*S. alba*	2	98	0	0	0	5/6	6/6	5–6	N	18
5	*S. aegyptiaca*	1	99	0	0	0	5/6	6/6	5/6	L	72
6	*S. euxina*	0	100	0	0	0	6/6	6/6	6/6	L	45
7	*S. alba*	3	94	3	0	0	4/6	7/7	4–7	S	75
8	*S. fragilis*	3	80	17	0	0	5/6	8/7	5–8	N	75
9	*S. fragilis*	1	76	23	0	0	6/6	8/7	6–8	L	51
10	*S. fragilis*	1	53	44	2	0	6/6	9/8	6–9	L	48
11	*S. fragilis* var. *decipiens*	3	50	33	14	0	4/6	9/8	4–9	N	30
12	*S*. *alba*	1	50	47	2	0	6/6	9/7	6–9	S	146
13	*S. alba*	0	98	2	0	0	6/6	7/6	6–7	S	70
14	*S. fragilis*	0	93	7	0	0	6/6	7/6	6–7	L	65
15	*S. alba*	0	92	8	0	0	6/6	7/6	6–7	A	18
16	*S. alba*	0	90	10	0	0	6/6	7/6	6–7	A	35
17	*S. fragilis*	0	90	10	0	0	6/6	7/6	6–7	L	92
18	*S. fragilis*	0	89	11	0	0	6/6	8/7	6–8	A	66
19	*S. alba*	0	89	11	0	0	6/6	8/7	6–8	S	106
20	*S. fragilis × S. alba*	0	64	36	0	0	6/6	8/7	6–8	S	40
21	*S. fragilis* (*×S. alba*?)	0	56	44	0	0	6/6	8/7	6–8	L	39
22	*S. alba*	0	50	50	0	0	6/6	8/8	6–8	N	20
23	*S. alba*	0	95	3	2	0	6/6	8/6	6–8	S	60
24	*S. alba*	0	92	6	2	0	6/6	8/6	6–8	S	85
25	*S. alba*	0	90	7	3	0	6/6	9/7	6–9	S	73
26	*S. alba*	0	79	18	3	0	6/6	8/7	6–8	A	14
27	*S. fragilis × S. alba*	0	76	23	1	0	6/6	8/7	6–8	S	40
28	*S. alba*	0	74	19	7	0	6/6	9/7	6–9	S	31
29	*S. fragilis*	0	70	26	4	0	6/6	8/8	6–8	S	73
30	*S. fragilis*	0	56	42	2	0	6/6	8/7	6–8	L	44
31	*S. alba*	0	56	39	5	0	6/6	9/7	6–9	N	97
32	*S. alba*	0	60	35	5	0	6/6	9/7	6–9	S	71
33	*S. fragilis*	0	38	52	10	0	6/7	10/8	6–10	L	54
34	*S. alba* var. *caerulea*	0	36	27	37	0	6/6	10/9	6–10	A	28
35	*S. fragilis*	0	34	41	25	0	6/6	10/9	6–10	N	71
36	*S. alba*	0	27	46	27	0	6/6	10/9	6–10	N	33
37	*S. alba*	0	25	59	16	0	6/7	10/9	6–10	N	65
38	*S. alba*	0	24	33	43	0	6/7	10/8	6–10	N	63
39	*S. fragilis*	0	23	67	10	0	6/7	10/9	6–10	N	73
40	*S. fragilis*	0	19	63	18	0	6/7	10/9	6–10	N	63
41	*S. fragilis*	0	15	62	23	0	6/8	10/9	6–10	L	41
42	*S. fragilis × S. alba*	0	13	84	3	0	6/7	10/8	6–10	A	91
43	*S. fragilis*	0	13	81	6	0	7/8	9/8	7–9	L	34
44	*S. alba × S. fragilis*	0	10	53	27	0	6/8	10/9	6–10	N	31
45	*S. fragilis*	0	4	65	31	0	7/8	10/9	7–10	S	70
46	*S. alba*	0	4	30	66	0	8/9	10/9	8–10	S	69
47	*S. fragilis*	0	3	41	56	0	8/9	10/9	8–10	A	43
48	*S. fragilis*	0	3	56	39	2	7/8	11/10	7–11	L	94
49	*S. euxina*	0	26	46	26	2	7/8	?	7–10	L	51
50	*S. fragilis*	0	19	54	23	4	7/8	10/9	7–10	L	39
51	*Salix*	0	0	30	70	0	8/9	10	8–10	S	15
52	*S. alba*	0	0	21	79	0	9	10	9–10	N	14
53	*Salix*	0	0	4	96	0	9	10	9–10	S	12
54	*Salix*	0	0	3	97	0	9	10	9–10	N	16
55	*S. alba*	0	0	25	74	1	8/9	11/10	8–11	S	55
56	*S. fragilis*	0	0	44	51	5	8/9	12/10	8–12	S	38
57	*S. alba*	0	0	14	79	7	9/9	11/10	9–11	N	22
58	*S. alba × S. fragilis*	0	0	10	88	2	8/9	11/10	8–11	S	26
59	*S. alba*	0	0	4	95	1	9/9	11/10	9–11	N	65
60	*S. alba*	0	0	2	84	14	9/10	12	9–12	A	58
61	*S. alba*	0	0	1	78	21	10/10	12/11	10–12	N	36
62	*S. alba*	0	0	4	46	50	10	12/11	10–12	N	12
63	*S. alba* var. *splendens*	0	0	0	93	7	10	12/11	10–12	N	14
64	*S. alba*	0	0	0	80	20	10	12	10–12	A	20
65	*S. alba*	0	0	0	71	29	10	12	10–12	N	12
66	*S. alba*	0	0	0	63	37	10	12	10–12	N	38

**Table 3 plants-12-00497-t003:** The ovule data for *Salix alba, S. fragilis* and *S excelsa* with 6 to 12 ovule/valve. Each capsule in *Salix* has 2 valves and the presented data include the percentage of valves in a catkin with different numbers of ovules. The additional characteristics are included: the number of ovules per ovary (min/often and max/often), the ovule indices (the minimum and maximum number of ovules per ovary), presence of trichomes at bract apex (A—bracts abscised; N—none; S—short), and number of studied ovaries. The entries with the specimens’ numbers (column 1) were colored according to their names based on their original identification—blue for *S. alba*, yellow for *S. fragilis*, white for *S. excelsa*—for visual presentation of the species’ distribution throughout the tables.

No	Specimen	Percentage of Valves with Each Ovule Number							No. of Ovules per Ovary		Ovule Index	Trichomes	No of Ovaries
		6	7	8	9	10	11	12	Min/ Often	Max/ Often			
67	*S. alba*	91	7	2	0	0	0	0	11/12	14	11–14	N	42
68	*S. alba*	2	5	78	15	0	0	0	13/16	17	13–17	A	48
69	*S. alba*	0	5	84	11	0	0	0	14/16	18	14–18	S	55
70	*S. alba*	0	24	71	5	0	0	0	15	17	15–17	N	19
71	*S. excelsa*	0	5	67	28	0	0	0	15	18	15–18	N	9
72	*S. excelsa*	0	0	14	73	9	3	1	16/17	21/18	16–21	N	54
73	*S. alba*	0	0	0	88	8	4	0	18	20	18–20	N	36
74	*S. alba*	0	0	0	62	20	9	9	18	21/19	18–21	A	17
75	*S. fragilis*	0	0	0	55	27	9	9	18/19	22/19	18–22	A	11
76	*S. alba*	0	0	0	0	26	48	26	20/21	24/23	20–24	A	27

## Data Availability

Data available in a publicly accessible repository at the Center for Open Science’s Open Science Framework (https://osf.io/kdqm5/).

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
