# Peer review of "The Ovule Number Variation Provides New Insights into Taxa Delimitation in Willows (Salix subgen. Salix; Salicaceae)"

_plants, 2023, doi:10.3390/plants12030497_

Round 1

Reviewer 1 Report

REVIEW

Variation of the ovule number in Salix babylonica (section Subalbae), S. alba and S. fragilis (section Salix) by A.N. Marchenko & Y. A. Kuzovkina

GENERAL REMARKS

Present manuscript deals with variation in number of ovules in willows (Salix, Salicaceae). This genus is well known for its complex taxonomy, major problems come from extensive hybridization, reticulate evolution, young evolutionary age of species, and lack of distinct morphologic characters on generative organs, combined with large phenotypic variation on vegetative organs. Species delimitations is thus often based on vague characters, different taxonomic approaches are controversial, and without knowledge of genetic relationships is often nearly impossible to clearly delimit boundaries not only between species and their hybrids, but also between some species.

This paper presents observations on number of ovules in some species of willows, and it reveals its successful application in species delimitation within tricky group of narrow-leaved taxa of Salix subgen. Salix. These observations deserve their publication in high-ranked (Q1) journal as Plants. However, it has several very weak parts that require considerable rewriting.

It is known for decades that number of ovules can be useful character in distinguishing some similar/related species. Authors did great job with comparing larger number of species and morphotypes and they provided the scientific community with new useful tools, which can be added to other methods used in willow research. They put the work of previous classical willow taxonomist on more solid backgrounds. However, authors over-valuated the importance of this single character and their conclusions that led to new nomenclatorial adjustments are beyond the real value of this single character. I am convinced that nomenclatorial adjustments should be based on broader studies, which include various methodological approaches and techniques, and not only on variation of single character.

The presented paper has its bright side on describing the variation of number of ovules. It is pity that authors divided their observations into several minor papers, I am convinced that their observations on ovule variations on broader species spectrum published without any doubtful nomenclatorial changes would attract attention of not only willow researchers, but might be very inspiring for researchers in other genera. However, also this part of broader study is worth publishing in high ranked journal like Plants. But before accepting the manuscript for publication, it requires considerable improvement.

Dark side of the manuscript are nomenclatorial adjustments, and structure of the paper. If these major issues are improved, I would be happy to see the data (manuscript) published!

My major concerns are described in following remarks:

NOMENCLATORIAL ISSUES

In past ca. 15 years, willow taxonomy experienced nomenclatorial disaster (e.g. Salix fragilis/euxina, Salix ×dasyclados, or more recently weeping willows), which caused instability of willow taxonomy. Revolutionary changes lead to new species concepts, which were consequently adopted, but recently it is more and more clear, that it was based on wrong expectation, wrong data and very speculative hypotheses, but lacking solid background (lack information on genetics, population variation, spatial distribution). I am afraid that proposed nomenclatorial adjustments could cause similar troubles as above-mentioned cases and will be another “nomenclatorial willow nightmare”, that it might have similar disintegrating impact and bring new instability in already fragile willow nomenclature. I therefore strongly suggest authors to responsibly considerate what really deserves new taxon/new combinations and for which taxa it is better wait until more data are available.

If authors decide to describe some of proposed nomenclatorial adjustments (which are, subjectively, eligible), they must prepare it in agreement and according to the Nomenclature code, i.e. the International Code of Nomenclature for algae, fungi, and plants – so called “Shenzen Code 2018” (see https://www.iapt-taxon.org/nomen/main.php). Nomenclature has strict rules and these must be followed in scientific papers.

Authors propose 12 nomenclatorial novelties. Some of them might be considered eligible, some are dubious (subjectively), and some are clearly wrong (objectively, violation of rules of the Nomenclature Code).

The most critical (weak) issues of proposed nomenclatorial changes are:

1) Variability of willows is rather complex, especially due to young evolutionary origin of species, limited genetic differentiation between some species, and enormous phenotypic plasticity, especially in tetraploid taxa (which are subject of presented study!). Therefore, establishing new varieties (almost solely) on number ovules is extreme shortcut. Taxa, for which are proposed new names/taxonomic ranks should be studied extensively in order to better understand what they really represent. I strongly suggest authors to avoid such approach (delimiting taxa on number of ovules only) – somebody can select any other character and describe another myriads of forms and varieties. And that is not science, that is philately, the approach of seeking differences between herbarium specimens ended in 19th century, modern biosystematics should include more information when describing new taxa,

2) Taxonomic rank of variety is usually considered for a infrataxon, which morphologically differs from type taxon, it is morphologically stable across its distribution area, based on genetic differentiation and distribution are such taxa described as subspecies, variety or forms. Forms usually represent interesting morphotypes, but regulated by simple Mendelian genetics, occurring randomly in distribution are of the species.

Taxa like Salix babylonica ‘Tortuosa’ or S. babylonica ‘Crispa’ represent such variation caused by simple Mendelian genetics (it was experimentally confirmed that twisted character of S. ‘Tortuosa’ has simple Mendelian genetics). These are artificially distributed taxa with unknown occurrence in nature. Apparently sometimes in the past such morphotypes were discovered in nature, and since then are reproduced just artificially. There is no need to describe such taxa at rank of form, the cultivar name clearly characterizes the taxon, these are not natural taxa!

Similar approach should be applied on Salix ‘Sacramento’ (Salix babylonica “var. sacramenta”) and Salix ‘Wisconsin’ (Salix babylonica “var. dolorosa”), especially if some authors hypothesize their hybrid origin! (unfortunately not confirmed by genetic analyses yet). Rank of variety seems to be rather inappropriate for such cultivars of unknown origin. These are artificial taxa and cultivar name is taxonomically the most appropriate!

3) Proposing new name for the most typical form of Salix babylonica is superfluous. Carl Linnaeus described Salix babylonica as weeping form (without any doubt) and that is the morphotype named by Santamour & McArdle as Salix babylonica ‘Babylon’. Therefore, taxonomically it is Salix babylonica var. babylonica and therefore, if name Salix babylonica var. babylon Marchenko is published, it immediately would fall within synonymy of S. babylonica var. babylonica.

4) Hybrid taxa can have name, but have to be written with hybrid formula (See Article H.3). It is impossible to describe hybrid taxa as Salix babylonica var. fragilis (because authors suspect that taxon have something from S. fragilis), or similarly S. alba var. fragilis or S. fragilis var. alba. Such taxa have already validly published names and if these nomenclatorial proposal are published, they immediately become illegitimate names as they are superfluous. Authors have to take into consideration that (especially tetraploid) hybrids are more variable than their parents!

To summarize these points, out of 12 new nomenclatorial proposals, 6 following taxa should be rejected from describing as new taxa, because if these new names are published, they would have to be considered (according to Nomenclature Code) as illegitimate names (superfluous names). Editorial board of section “Plant Systematics, Taxonomy, Nomenclature and Classification” should reject any attempt to publish illegitimate taxa!

a)     Salix babylonica var. babylon – belong to the type variety Salix babylonica var. babylonica. In order to highlight its weeping character, cultivar name can be used (S. babylonica ‘Babylon’).

b)    Salix babylonica var. fragilis – authors expect “greater involvement” of Salix fragilis in Salix babylonica, this means they expect hybrid origin. Such hybrid was already (several times) described:  as S. ×pendulina Wender. = S. ×blanda Andersson = S. ×elegantissima Koch

c)     Salix babylonica var. alba – authors indicate involvement of S. alba in S. babylonica genome, such hybrid was repeatedly described and although the nomenclature of such hybrids is not clear yet, some of published names probably belong to such hybrid combination, i.e. S. ×salamonii Carr. = S. ×sepulcralis Simonk. = S. ×chrysocoma Dode, at rank of variety is valid name Salix alba var. vitellina-tristis Ser.

d)    Salix fragilis var. babylonica – intermediate between S. fragilis and S. babylonica suggests it might be hybrid between those species, i.e. repeatedly described in past, as S. ×pendulina Wender. = S. ×blanda Andersson = S. ×elegantissima Koch

e)     Salix alba var. fragilis

f)     and Salix fragilis var. alba are both expected by authors to be of hybrid origin, therefore they fall within validly described Salix ×rubens Schrank.

Four new names and combinations seem be to inappropriate at rank of form or variety, because they represent just artificial taxa, cultivars. But apparently, publishing them is not violation of rules of the Shenzen Code.

a)     Salix babylonica var. sacramenta

b)    and Salix babylonica var. dolorosa – authors revealed it has unique number of ovules, however authors do not present any other argument that such artificial cultivars deserve rank of new variety! Although their origin is unknown, it is generally expected to be of hybrid origin, therefore genetic analyses explaining their origin are required. Describing these cultivars as new varieties solely on number of ovules is, in my humble opinion, too preliminary.

c)     Salix babylonica var. matsudana f. annularis

d)    and Salix babylonica var. matsudana f. tortuosa are artificial cultivars, however morphologically very distinct (and beautiful), they better should be treated as cultivars only (S. babylonica ’Tortuosa’ and S. babylonica ’Crispa’). Furthermore, Salix babylonica f. tortuosa is already validly described! (Salix babylonica f. tortuosa Y.L.Chou in Bull. Bot. Res., Harbin 1(1-2): 159 (1981)). If authors decide to make new status (with var. matsudana), authors must provide solid evidence that name “var. matsudana” has priority against “var. pekinensis” and authors must study the Code in order to make correct selection of the basionym! I cannot do it instead of authors (it would take another hours of searching), I have doubts they choose the right basionym! This must be considerably evaluated and clearly explained.

Remaining two new combinations are taxonomically not clear and ambiguous

a)     Salix alba var. excelsa (S.G.Gmel.) Marchenko – Salix excelsa is widely distributed taxon. It is still not solved yet, what is the true relationship among S. alba, S. fragilis, S. australior and S. excelsa. However, making new status as variety under S. alba (and giving this species same rank as some artificial cultivars) based only on number of ovules, is in my humble opinion, extreme taxonomic shortcut. Such radical changes should not be done without proper knowledge of taxon. This, unfortunately, reminds me impetuous nomenclatorial changes done in S. fragilis/euxina or S. ×dasyclados in past yearsI urge author to be responsible.

b)    S. babylonica var. matsudana f. umbraculifera – it is not very clear, what is the origin of taxon. Do authors believe it is hybrid between S. babylonica and S. fragilis? If so, then it should be considered an infrataxon within S. ×pendulina.

However, if authors want to describe taxa S. babylonica var. matsudana f. umbraculifera, Salix alba var. excelsa, Salix babylonica var. matsudana f. tortuosa, Salix babylonica var. matsudana f. annularis, Salix babylonica var. dolorosa and Salix babylonica var. sacramenta, there are no limitations and names can be validly described, if authors wish to do so.

But if authors want to validly publish new combinations and new varieties, they have to follow rules and recommendations of the Nomenclature code!

In present manuscript are following violations of rules of the Shenzen code:

a)     Any of the new taxa has clearly established type specimen! Authors probably consider Table 2 as summary of types, but it is not clear enough. Authors should clearly state for all new taxa the HOLOTYPUS. For new combinations of already described taxa should be stated the type of the basionym. Especially for Salix “matsudana” cultivars should be carefully selected appropriate basionym type (and not some random sample from some park collected in 20th century). If NEOTYPUS has to be established, then it must be clearly explained why.

I recommend to write for each new taxon or new combination or status separate paragraph, in which are all required information clearly presented: diagnostic characters, any other known information, known distribution (if any), explicitly established type specimen (and clearly established what type of specimen) with link to image of the Type (present type specimens of all new taxa in the paper!). For new combinations must be explicitly mentioned the basionym (otherwise it will not be valid nomenclatorial change!). If types are not properly selected, new taxa and new combinations might be considered as illegitimate name (nomen nudum).

b)    It is not obligatory, but it is usual (and good) practice to present image of type specimen of new taxa (combinations) in the paper. To show readers, what the new taxon really is. If authors do not want to clearly show, on which type specimen is based their new taxa and combinations, I would suggest reject such paper, because such nomenclatorial adjustments only complicates willow systematics. Especially in situation, when types are deposited in collections not easily accessible to scientists, such as MHA, CONN, etc. I suggest authors to clearly show, how plants on type specimens look, to scientific community, what proposed taxa really are. Soon or later, genetic data will be available, and it will be crucial to clearly known, on what proposed new taxa are really based. Knowing just number of ovules (and some few details) is hardly insufficient for understanding proposed taxa. 

c)     Authors should correctly describe their nomenclatorial adjustments. Proposed taxa  S. babylonica L. var. sacramenta Ðœarchenko, comb. et stat. nov.” and “S. babylonica L. var. babylon Ðœarchenko, comb. et stat. nov.“ cannot be considered neither “comb. nov.”, nor “stat. nov.” (mentioning both has no logic), because it is not based on previously validly published basionym – see Article 6.10. These are both “var. nov.”

According to same Article (6.10), forms of S. “matsudana” should be stated as “stat. nov.” only, S. alba var. excelsa as “comb. nov.”

MANUSCRIPT STRUCTURE

The most important outcome, variation in number of ovules in willow capsules, is somehow hidden in the paper. As it is written, it is too (I would even say extremely) narrowly focused on willow readership. However, it has potential to catch interest of other scientists and to be inspirative for research in other genera, which have limited generative characters.

I therefore recommend rewriting the paper in order to be more open to other readers than willow geeks.

Presented manuscript has also bad structure of presented data, confusing tables and inappropriate division of paper.

It can be summarized in following remarks:

1)    Presented manuscript does not have logic structure: Besides usual chapters “Introduction” and “Materials and Methods”, it contains chapters “2. Results and Discussion” and “3. Discussion”. I strongly suggest changing the structure of the paper and use standard chapters “2. Results” and “3. Discussion”.  The chaotic chapter “2. Results and Discussion” requires considerable rewriting (including different arranging of tables).

2)    Title: standard practise for systematic works is to mention family of studied taxa. Here it is missing, and authors mention sections instead. However, using sections is not very useful (even for specialists it now little bit controversial, as it is obvious that old sectional concepts are not correct, but new concept requires further genetic studies). For non-willow readers sectional belongings provides any information

I recommend to use less  “controversial” title, which be more open to broader readership (and not only for willow “guys”). For instance: “Variation of the ovule number in willows (Salix babylonica, S. alba and S. fragilis; Salicaceae)”
or even better it would be if authors focus on highlights of the paper, for instance: “The ovule number variation provides new insights into taxa delimitation in willows (Salix subgen. Salix; Salicaceae).”

3)    Address: it seems there is error in address #1 (it seems that Moscow, Russia is doubled in the address, e-mail address is mentioned in inappropriate position).

Address 1 is mentioned as: “Russian Park of Water Gardens, Moscow, Russia; [email protected]Polevaya Str., 12, Klyazma mkr, Pushkino, Moscow Region, Russia

Probably it should correctly written as: “Russian Park of Water Gardens, Polevaya Str. 12, Mkr. Klyazma, ZIP-code missing Pushkino (Moscow), Russia; [email protected]”?

The journal Plants use the PubMed/MEDLINE standard format is used for affiliations and it should be corrected accordingly.

4)    Keywords: selection of key words is very unusual. Keywords must highlight major topics of the paper, but here are listed only species name, and furthermore in a nomenclatorial form not used in the paper! Why authors use as keyword “Salix matsudana”, but in the paper they use “Salix babylonica var. matsudana”? Same it is with other taxa.

I recommend changing keywords in order to get broader readership! I recommend using e.g.: number of ovules, willow, micromorphology, taxonomy, Salicaceae

5)    Table captions and Figure legends should be self-explanatory, always.

Caption of Table 1 lacks important information. Firstly, it is not clear enough what it is “Percentage of valves with each ovule number”. It will be very useful to learn readers that each capsule in willow has 2 valves and clearly explain that it presents data on number of ovules per each valve (out of 2). Please, explain it well, what is presented here. It must be explained in Table caption.

Secondly, in Table 1 captions cannot be written “some morphological characters in 30 specimens of S. babylonica”, but clearly explained which morphological characters! In table head could be the character shortened (in order to avoid unpleasant dividing of words into 2-4 rows), but well explained in caption.

Explanation for abbreviations (L, S, N, A, P, G) should be well explained in caption too. Leaving it at very end of enormously long table also makes confusions.

Caption for Table 2
is not self-explanatory too. Head or caption should include information that it is proposed new taxon name. Further it should be explained, what is meant by type in the Table! I would expect it is some kind of type herbarium specimen required by the Nomenclature code (Article 8). However, it is nowhere in the manuscript mentioned anything about Holotype, Lectotype, Neotype or other kind of types in text. Which in strong controversy to the Nomenclature code! If it is really meant as “type specimen”, it must be clearly stated together with each proposal for new name (new combination) and must be clearly state d what kind of type material the herbarium specimen represents!

Column “Description” is also not clear enough. Did authors mean diagnostic characters for new taxa?

Including synonyms for some of presented taxa makes the table uneasy to read. It should be discussed somewhere else.

Generally, Table 2 seems to me very confusing and probably it would be better to make special paragraph for each new taxon/combination and discard such table from manuscript.

Table 3 has same problems as Table 1. Explanation for first 6 columns is little bit clearer “…with 1 to 6 ovules/valve”. It will be very useful for readers to well explain it (whether it really means number of ovules per valve).

Figures are fine. Plants is online journal; therefore I would appreciate to see even more images of ovules, studied plants, and especially all type specimens of newly described/combined taxa!

In Figure 2, it would be very useful to add comparison with leaf-margin and bud of S. babylonica var. babylonica (S. ‘Babylon’) and S. babylonica var. dolorosa (S. ‘Wisconsin). Similarly in figure 6 would be extremely helpful to see other stipules, i.e. of the ‘Wisconsin’ and ‘Sacramento’ morphotypes.

In Figure 3, it is not clear what the specimen represents. Is it considered to be a type of (illegitimate) new variety var. babylon? Figure legends must self-explanatory too.

6)    Introduction: It starts directly with describing one of studied species. I strongly recommend to include “more introductory” information, such as what are problems causing taxonomic complexity in willows, what are (superficially) relationships among willows in world and explain why it is so useful to test another “tool” for species identification. Then can be introduced studied species…

The used terminology has to be also well explained, I expect that nobody besides willow community knows that each capsule contains two valves, in which ovules can be counted.

7)    Material and Methods: I recommend including list of studied samples, as own Table. Because when each specimen is mentioned in results (Table 1 and Table 3), it cause unpleasant and visually confusing large tables with very broad rows. If all studied specimens are presented in own Table, locality can have broad columns and will not damage the table. In results can be each particular specimens referenced by number (as it is already used in Table 1 and Table 3). Information on Plant material usually belong to MM section. In present form, when it is placed in Tables in results, it is unusual and it is causing problems with its reading.

8)    Results: I strongly suggest authors to focus only on “raw” data. To present variation in number of ovules, in clear synoptic Tables, exclude origin of specimens and focus on taxa and obtained data. Authors should describe obtained data (and not to resign on something “Table 1 presents a snapshot”. Make some summary, which can help to better orientate in Tables. Avoid taxonomic discussion in Results section.

9)    In Discussion, discuss pros and cons of the method, discuss taxonomic consequences, if you wish, propose nomenclatorial adjustments.

10) In Conclusion, just summarize the major outcome and benefits for the scientific community… The present Conclusions can be shortened to 1/3 (at least) and partly moved to Discussion. Describing nothotaxa as varieties of parental species is illegitimate and should be avoided in Discussion/Conclusions, Figure 7 requires correction, taxonomically it is incorrect, but for presenting number of ovules it is useful!

11) Citations and References: The journal’s citation policy is not accepted by authors in presented manuscript. Citations are mentioned in a form “Author, year” in manuscript body, but in the References listed as numbered (but ordered alphabetically!). References therefore are not formatted to MDPI style! It is rather confusing and difficult to follow. Authors should carefully follow the Instructions for Authors (https://www.mdpi.com/journal/plants/instructions#preparation) and adjust citations in manuscript and reference list accordingly!

Plants accepts “free format submission”, but after revision it is expected to change formatting to the MDPI policy, revised version should adopt above mentioned formatting.

OTHER MINOR MISTAKES AND ERORRS

-     Line 34: “piedmont Kirghizia” – correct country name is “Kyrgyzstan” à change to “lowlands of Kyrgyzstan” or simply “Kyrgyzstan”.

-     Lines 40-41: “The delimitation of these species is rather difficult as both species belong to the section Salix and are morphologically similar” – it is not true and cited references do not support this information. Salix alba and Salix fragilis are clearly distinguished taxa, genetically well separated. Problem is caused by intensive hybridization, which blurs the species limits, but species themselves are clear taxa!

-     Lines 44-45: “Still, the boundaries between these species are defined by relatively few diagnostic characters and unambiguous field identification is challenging”. – be very careful with such dtrong statements. It is not true. It is very difficult if you study only herbarium specimens, but if you have good field-observation background, you easily recognize.

-     Page 7, 3rd and 4th row, 2nd column: typing error “Koizumi”

-     Lines 348-350: “a hybrid of S. babylonica var. babylon and S. alba var. fragilis), which was discussed in the study related to the hybrids of S. babylonica (Marchenko, 2019; Kuzovkina, 2022)” – this should be better explained, because cited works are unavailable to broader readership, these publications are not indexed by any major scientific database!

-     Lines 525-542: Authors hardly overestimate the importance of their method. They have to keep in mind, that they did not provide any prove that their data are completely correct. They simply classify taxa based on ovule number, but they are not sure if these are truly the taxa. Additionally, they argue that erroneous determination can bias molecular studies. That is not true, if reliable markers (and especially in recent years using NGS approaches) are employed, it has power to detect false determination! It is right that ovule number might be useful tool for determination, but without knowing other morphologic characters (and population variation, which is often lacking in herbarium-based studies!) is the method blind. Counting the ovule number is very useful tool, one among other ones, which definitively can help the willow community in solving taxonomic puzzles.

-     Lines 544-554 (paragraph 3.2.4) it is useful to provide information for Salix tetraspema and Salix mucronate. However, these taxa are hardly not cultivated in Europe (and I guess neither in US). Therefore, I will appreciate to include information on Salix humboldtiana, which is also superficially similar to S. babylonica and is quite often cultivated in Europe (Mediterranean) and Latin America. It is often incorrectly identified as S. babylonica in both regions, it would very helpful to known ovule number for this species.

In Olomouc, November 22, 2022.

Radim J. Vašut, Associate Professor of Forest Botany, Palacký University in Olomouc.

Reviewer 2 Report

The topic of the article is interesting and relevant both for the knowledge of this group of plants and for the assessment of their potential productivity. Unfortunately, despite its strengths, the paper has critical shortcomings that require a major revision. For this reason, the following review will focus on the most important observations.

1. The authors propose a number of new nomenclatural combinations, but these would be invalid if published as currently presented in the article, because they are in conflict with the provisions of one or more of the articles of the International Code of Nomenclature for Algae, Fungi, and Plants (ICNAFP 2018). All the new combinations are contrary to the provisions of Article 41.

2. The abstract needs to be expanded and restructured, as it does not give an impression of the content of the article as a whole.  Keywords do not meet the requirements. They must be replaced by real keywords.

3. In my opinion, the introduction needs to be supplemented, especially the beginning, on the importance of ovule number not only in Salix taxonomy but also in the delimitation of other plant groups. It is also worth at least briefly discussing another important issue related to ovules, the potential fecundity. 

4. I consider it a very serious shortcoming that there is no statistical analysis of the data obtained. In the section of results, only the pure values are presented, but there is no assessment of whether there are significant differences between them or whether this is just a subjective impression. The fact that the tables are huge and contain information that is not necessary for the results section also makes it difficult to understand the results. The origin of the material should be in the methodology section. Furthermore, I suggest that the methodology section be placed before the results, as it is done in some of the articles published in this journal. 

5. The new taxonomic combinations should not be in the table but in the text and in accordance with the requirements of the Code (see Note 1). Furthermore, there is no clear justification for the taxonomic decisions made. I have read the article, but the authors are not convinced that the proposed nomenclatural changes are necessary. 

6. How are the proposed new nomenclatural combinations, where the name of a cultivar is replaced by the name of an infraspecific taxon, compatible with the Code of Nomenclature for Cultivated Plants? 

7. In my opinion, some of the proposed new combinations are illlegitimate (e.g.: S. fragilis L. var. babylonica Marchenko, comb. et stat. nov.; S. fragilis L. var. alba Marchenko, comb. et stat. nov.; S. alba L. var. fragilis Marchenko, comb. et stat. nov., etc.). The epithets of the varieties correspond to those of other previously published species of the same genus (see Art. 23, 24, 53 of ICNAFP, 2018). 

8. I think that the methodology should be moved to the beginning of the paper after the introduction and significantly supplemented. The methodology is now based on references, and the subsection "Ovule count" describes in detail the order in which the specimens are arranged in the tables rather than the procedure itself. 

Reviewer 3 Report

The authors expose the utility of the number of ovules for a complex genus and its remarkable utility to separate and identify different species of this genus. This feature should be added to the descriptions and further taxonomic studies of this genus. However, some important flaws are found in the paper, and I would recommend to resolve them to get the paper published.

General comments

  • The nomenclature of the many infrageneric and infraspecific taxonomic names of Salix are not properly written accordingly to ICN rules. The use of italics should be used for any infrageneric name.Review the whole manuscript (including the title).  
  • The authority of the taxa should be written only the first time the taxa is written in the main text (introduction, material and methods, results and discussion). Do not add the author of the taxa in the abstract. 
  • Review the address of the authors, something strange appears in the middle of the first address. 
  • line 60, add ‘and’ between the two surnames.
  • I would recommend to write the results and the discussion as two separated sections. 

Tilte

The word ‘and’ should not be in italics. I would recommend to modify the title to include the main proposal of the study.   

Keywords

Review the proper selection of the keywords. It is a current list of taxa names and in my opinion, it is not totally adequate.  

Introduction

  • I recommend to introduce the complexity of the genus (morphological and phylogenetic data), the followed taxonomic proposal of thus genus, and after that the studied species and their complexity. More detailed data about the complex use of vegetative and reproductive features to identify species of Salix should be also included. 
  • Do the authors follow a taxonomical classification based on only sections (e.g. Fang et al., 1999) or subgenera and sections (e.g. Argus, 1997)? Add information about the followed taxonomical classification, clarify which one is followed in the paper. 
  • Recently, Salix fragilis was considered as a species not a hybrid by these authors. Some additional information about it sine no data is included in the introduction.  
  • lines 60-62: How do you support this statement about the number of ovules is a stable and consistent morphological feature? No previous detailed works are cited. In addition for S. alba, the number of ovules is not so consistent, since certain variability was detected. 

- Clarify the main objetives of the study.  

Material and methods

  • The total studied n is missed for each species and specimen. Specific data should be included in the text. 
  • Which herbaria were consulted? Where are the vouchers kept? No mention is done. 
  • The taxonomic identification of the studied vouchers is only based on the previous identification? No additional data is given about other morphological features typical for each species and for each specimen. Have the authors revised the previous taxonomic identification? It is rather important to give additional data about the identification of the vouchers, and not only based on the number of ovules per valve and per ovary.   
  • In the tables, the meaning of some data are not adequate explained (e.g. CONN, MHA, color of the cells, number of ovaries). 
  • The explanation about the selection of these three species should be well-supported. No other Salix species could be misidentified for any of the here selected species? The species selected belong to two different sections, and are these two section well-differentiated based on morphological features? In some parts of the study, a hybrid was studied, but not described methodology is given for them. 
  • No explanation is given about the taxa S. mucronata and S. tetrasperma. Why are they selected? The inclusion of these data should be also included in material and methods, and also the explanation of their final selection. How many samples are studied? 
  • The given data is based on minimum/often and maximum/often. An explanation about the use of the “often” should be given, since no specific reason is given about what “often” is really considered. How do the authors define “often” to be considered for this and further studies?
  • The additional used characters for each species are not identical, since the presence/lack of pubescence of the ovary is used for S. babylonica, and the length of the trichomes for S. alba and S. fragilis. 
  • Lines 577-590, this part of the m&m should be changed to results. They are not purely m&m. 
  • A statistical part should be added to support the separation of these three species, including some ANOVA and PCA analyses.        

Results and discussion

  • I recommend to write a taxonomical proposal subsection to include the list of the new proposed taxonomic combinations. Are these taxonomic combinations based only on one specimen? The ICN rules should be used to propose the new combinations since some of them are not adequately described and hence, it is not a valid taxon  (e.g. no type data is given). A detailed revision must be done. 
  • The origin of the samples might be included in a separate table, and the herbarium where the material is kept should be clearly indicated. 
  • The use of the number of ovules per valve shaped be used in combination of other morphological features that support their identification (especially on those male specimens or samples without inflorescences). This is also important when the authors state the existence of a transition between species. 
  • Part of the conclusions correspond to hypothetical ideas without enough support, and other data from other disciplines would be used to sustain the proposed ideas.This should be considered, for example, for the relationships between the climatic conditions and the evolutionary processes of this group. Which geographic areas and samples were used to give this hypothesis? In addition, part of the conclusions would depart pf the discussion. The conclusion section is too long. 
  • Despite the potential use of the number of ovules per valve, the given new combinations would be supported by additional morphological data, and I would recommend the inclusion of an identification key to facilitate the identification of the studied group.
  • The given explanations about introgression and hybridization should be supported by morphologic and phylogenetic data. Similarly to the hypothesis about the evolution progress from S. babylonica and S. fragilis to S. alba. Are there any phylogenetic data that support this statement? How do you know that a low number of ovules correspond to a ‘old’ feature within this genus? Have you traced this feature along a phylogeny of the genus Salix?    

Reviewer 4 Report

The article brings valuable new knowledge about importance of minimum/maximum number of ovules per valve and per ovary for the identification of morphologically variable and similar species Salix babylonica, S. alba and S. fragilis. The aim of this work well-defined and brings original results. The paper is an extension of previous research in this area and it is evident that the authors are well acquainted with the issue. Cited references are relevant and many of them are recent publications (within the last 10 years). An exhaustive research was carried out and numerous conclusions were presented that can help in the determination of individuals within the genus Salix. In addition, several proposals for defining new varieties within the S. babylonica, S. alba and S. fragilis species were presented. The tables as well as the figures are well done and of high quality.

The main objection concerns the structure of the article. Personally, I prefer Materials and Methods before the Results chapter, because that way the procedure and expected results are explained to the reader in advance, which makes it easier to follow the obtained results. In addition, there is a chapter in the paper called Results and Discussion and the chapter Discussion, which is completely unnecessary. Since the article contains a lot of measured results and data, as well as written comparisons and discussions, the reader needs exceptional concentration to follow the text. For the clarity of the article, I think it would be vise to place Tables 1-4 to the Appendix. I strongly suggest that the text from the Discussion chapter gets transferred to the table format and moves to the Results and Discussion, at the end of each small chapter on an individual group (2.2.1, 2.2.2,...). That might need some adjustement of the Result text (maybe some of explanations can be reduced, which would help readers to stay focused). Another option is that the current Discussion (with some minor adjustments) becomes the Conclusions chapter. In my opinion, the current chapter Conclusions is too long and does not present precisely and concisely, which are the main conclusions of the conducted research.

(Please check again all the latain names, some of them are not Italic written.)

Round 2

Reviewer 1 Report

Revised version of the manuscript is considerably improved. I appreciate that the most critical issues were corrected, controversies omitted and explanatory notes added. Solving the controversies against ICBN with the non-nomenclatural designations as groups is the right way, in my opinion. Further investigations might shed light on what these taxa taxonomically represent.

I recommend the manuscript for publication after correcting minor issues.

MINOR REMARKS

-          Authors’ decision to avoid describing intermediates as new varieties is commendable. However, I recommend to name intermediate groups completely, i.e. not just as “Babylonica-fragilis – group”, but “S. babylonica-fragilis group”, because other taxa are referred to as (e.g.) “The pure S. fragilis”. All (including these intermediates) are Salix taxa. (I recommend to correct it in whole text + Figure 7).

-          Salix babylonica var. sacramenta – authors should explain at first mention (page 7) what they consider under this name. If authors consider this taxon as variety, but they are not going describe it in this paper, I recommend to mention it as “S. babylonica var. sacramenta Marchenko et Kuzovkina, ined.” (just to clearly express they are considering it as such), or use the cultivar name (as used in Santamour and McArdle), i.e. S. babylonica ‘Sacramenta’ (to be in agreement with nowadays practise). In its present form it is nomenclatorially incorrect.

-          P. 8, line 696 – I recommend to include synonymy at first mention of this cultivars, i.e. (syn.: S. babylonica ‘Crispa‘)

-          P.19. lines 2054-onwards: some mistake in numbering of references

Reviewer 2 Report

The quality of the updated version of the article has improved considerably. I would like to make a few comments and suggestions.

1. I recommend that the tables with information on the herbarium specimens studied should not be placed as supplementary material, but should form two appendices. Supplementary material often loses its relevance when detached from the text of the article.

2. The text is riddled with technical errors (unnecessary full stops, missing spaces between words, extra spaces, etc.). The text of the article needs to be carefully read and edited.

3. Some of the references are not numbered in the reference list.

4. In a previous review I wrote about statistical processing of data. I realise that this is a complicated job because of the very heterogeneous samples. Nevertheless, the statistical analysis would show the real differences between the different Salix taxa in terms of number of ovules. I believe that statistical analysis can be dispensed with in this article, but statistical analysis must be used to assess the significance and reliability of the method. 
